# Age-Dependent Effects of Oxytocin and Oxytocin Receptor Antagonists on Bladder Contractions: Implications for the Treatment of Overactive Bladder Syndrome

**DOI:** 10.3390/biomedicines12030674

**Published:** 2024-03-18

**Authors:** Masroor Badshah, Jibriil Ibrahim, Nguok Su, Penny Whiley, Ralf Middendorff, Michael Whittaker, Betty Exintaris

**Affiliations:** 1Hudson Institute of Medical Research, Monash University, Clayton, VIC 3168, Australia; penny.whiley@hudson.org.au; 2Monash Institute of Pharmaceutical Sciences, Parkville, VIC 3052, Australia; jibriil@gmail.com (J.I.); eunice.su@monash.edu (N.S.); michael.whittaker@monash.edu (M.W.); 3Institute of Anatomy and Cell Biology, Justus-Liebig-University, 35390 Giessen, Germany; ralf.middendorff@anatomie.med.uni-giessen.de

**Keywords:** overactive bladder, lower urinary tract symptoms, oxytocin, oxytocin receptor antagonist

## Abstract

Overactive bladder (OAB) is an age-related disorder characterised by unstable bladder contractions resulting in disruptive lower urinary tract symptoms (LUTS), thus creating a profound impact on an individual’s quality of life. The development of LUTS may be linked to the overexpression of oxytocin receptors (OXTRs) within the bladder detrusor muscle, resulting in increased baseline myogenic tone. Thus, it is hypothesised that targeting OXTRs within the bladder using oxytocin antagonists may attenuate myogenic tone within the bladder, thereby providing a new therapeutic avenue for treating OAB. Organ bath contractility and immunohistochemistry techniques were conducted on bladder tissue sourced from young rats (7–8 weeks and 10–12 weeks) and older rats (4–5 months and 7–9 months). Organ bath studies revealed that oxytocin (OT) significantly increased bladder contractions, which were significantly attenuated by [β-Mercapto-β,β-cyclopentamethylenepropionyl1, O-Me-Tyr2, Orn8]-Oxytocin) (1 µM) (**** *p* < 0.0001) and atosiban (10 µM) in both young and older rats (** *p* < 0.01); in contrast, cligosiban (1 µM and 10 µM) did not inhibit OT-induced contractions in both young and older rats (*p* ≥ 0.05). Interestingly, cligosiban (1 µM and 10 µM) significantly reduced the frequency of spontaneous contractions within the bladder of both young (*** *p* < 0.001) and older rats (**** *p* < 0.0001), while atosiban (10 µM) only demonstrated this effect in older rats (** *p* < 0.01). Furthermore, immunohistochemistry (IHC) analysis revealed significant colocalization of nuclear-specific oxytocin receptors (OXTRs) in the contractile (smooth muscle) cells within young (** *p* < 0.01) and older rats (* *p* < 0.05), indicating OT may be a key modulator of bladder contractility.

## 1. Introduction

Overactive bladder (OAB) is defined as urinary urgency with frequency, nocturia, with or without urge incontinence [1,2]. Its overall prevalence is around 10 to 16.6%, affecting both men and women over 40, which worsens with age [3]. This may be because with advancing age, there is a deterioration of bladder function, such as a decrease in bladder capacity, an increase in detrusor instability, and low compliance with drugs [4,5]. The precise cause of OAB is unknown. One of the most likely causes is detrusor hyperactivity (DH), which could be either neurogenic or non-neurogenic in nature. Neurogenic detrusor hyperactivity (NDH) is associated with spinal cord injury, degenerative neuropathies, and inadequate cortical inhibition. Conversely, factors such as bladder ageing, bladder outlet obstruction, and chronic bladder irritation due to urinary tract infection (UTI), tumours, and stones contribute to non-neurogenic detrusor hyperactivity (NNDH) [6].

Several options are considered for the treatment of OAB. The first line of treatment includes biofeedback (to educate people on how to control the bladder and muscles that are engaged during urination), pelvic floor muscle training, and dietary changes [7]. If the first line of treatment fails, then there are certain invasive procedures, such as bladder reconstruction surgery, the use of botulinum toxin type A, and sacral neuromodulation [8]. The pharmacological treatment is divided into the following categories: antimuscarinic, beta-3 agonist, desmopressin, hormone replacement therapy, and tricyclic antidepressants. Among these, antimuscarinics are widely used as the drug of choice for the treatment of OAB [9]. M_2_ and M_3_ receptors are widely expressed within the bladder, the predominant being the M_2_ receptors [10]. M_3_ has a direct effect via phosphoinositide hydrolysis, while M_2_ acts indirectly via inhibition of adenyl cyclase and sympathetically mediated relaxation, i.e., beta-adrenoceptor [11]. The rationale behind treating OAB with antimuscarinic therapeutics is that it acts on the muscarinic receptors, particularly the M_2_ and M_3_ subtypes, and thus they inhibit detrusor contractions [12]. It has also been demonstrated that the urothelium and detrusor contain β-1, -2, and -3 adrenoreceptors. These receptors (in humans, in particular, the β-3 receptors) are triggered to cause G protein and adenyl cyclase activation, which raises cyclic AMP levels and relaxes the detrusor [13]. In 2013, a new beta-adrenergic drug, mirabegron, was introduced to act directly on beta-3 receptors present in the smooth muscle resulting in bladder relaxation [14,15,16]. Later on, in September 2018, another beta-3 agonist called vibegron, was approved in Japan for overactive bladder syndrome (OAB) treatment [17]. The main advantage of using vibegron is that there is a low risk of drug interactions as it is less likely to be metabolised by cytochrome P450 3 A4 (CYP_3_A_4_) or cytochrome P450 Family 2 Subfamily D Member 6 (CYP_2_D_6_) [18]. However, compliance with these medications is low because of their side effects, such as dry mouth, constipation, blurred vision, confusion, somnolence, increased heart rate, and dry eyes [12]. For this reason, numerous clinical and experimental studies have been conducted since this time to improve clinical outcomes and reduce the side effects of medical treatment [19,20,21].

A novel paracrine target that warrants attention is the hormone oxytocin (OT). OT is a hormone synthesised primarily within the magnocellular neurons of the paraventricular and supraoptic nuclei of the hypothalamus, which is then stored and released from the posterior pituitary gland into the systemic circulation. OT binds with a G-protein-coupled OT receptor, leading to the activation of the Gq protein, which then results in stimulation of phospholipase C (PLC). This PLC then cleaves phosphoinositol diphosphate into diacylglycerol (DAG) and inositol triphosphate (IP_3_). DAG then triggers protein kinase C (PKC), whereas IP_3_ causes the release of calcium from the sarcoplasmic reticulum. Finally, phosphorylation of proteins on the cell membrane occurs, leading to a rise in the intracellular calcium level [22,23,24,25]. This results in multiple organ functions, including (i) contraction of the uterus during labour [26]; (ii) contraction of the myoepithelial cells within the female breast during lactation [27,28,29]; and (iii) stimulation and contraction of the seminiferous tubules within the testis [30,31]. Previous studies also demonstrated the role of oxytocin within the prostate via facilitating smooth muscle contractions within humans, dogs, guinea pigs, and rats [32,33,34]. Some studies also reported the presence of OT receptors within the prostate of species such as humans, cows, rabbits, and sheep [26,28,29]. Interestingly, recent studies also highlighted the existence of the OT receptor gene within the bladder [31]. Thus, OT acts on oxytocin receptors and facilitates detrusor contraction. Hence, targeting oxytocin receptors utilising oxytocin receptor antagonists might offer new insight into the treatment of OAB [35].

Therefore, the main aims of the present study were to (i) examine the effects of oxytocin (OT) on bladder contractions; (ii) investigate the effect of OT receptor antagonists (atosiban, cligosiban and [β-Mercapto-β,β-cyclopentamethylenepropionyl1, O-Me-Tyr2, Orn8]-Oxytocin (ßMßßC) on spontaneous as well as OT-induced contractions in two different age groups of male rats, i.e., [young (7–8 weeks and 10–12 weeks) and older (4–5 months and 7–9 months); (iii) qualitative assessment of the oxytocin receptor expression within the bladder of young rats (7–8 weeks) and older rats (16 weeks); and (iv) characterise vasopressin (VP)-induced cross-over changes in response to OT within the bladder of young rats (10–12 weeks) and older rats (7–9 months).

## 2. Materials and Methods

### 2.1. Animal Ethics

The bladder tissue from different age groups of Sprague Dawley (SD) male rats was received from the animal facility located at Monash Institute of Pharmaceutical Sciences (MIPS), Parkville and Monash Animal research Platform (MARP), Clayton, Australia. All the procedures were carried out in accordance with the guidelines that are required for animal care and approved by the Ethics Committee (MIPS- 26791 and MARP- 00000).

### 2.2. Tissue Collection

Different age group of rats were euthanised by placing them in a CO_2_ chamber. An incision in the abdomen was made along the midline to expose the urinary bladder. The bladder was dissected into two equally weighed perpendicular strips and then placed in a specimen jar containing MEM (Thermo Fisher Scientific, Waltham, MA, USA).

### 2.3. Reagent Preparation

Different powdered reagents such as oxytocin (Sigma Aldrich, St. Louis, MO, USA), vasopressin (ProSpec-Tany, Ness-Ziona, Israel), atosiban (Sigma Aldrich, St. Louis, MO, USA), and ß-Mercapto-ß, ß-cyclopentamethylenepropionyl (Sigma Aldrich, St. Louis, MO, USA) were used. These reagents were first dissolved in mili-Q water, except cligosiban (MedChem Express, St. Lucia, QLD, Australia), which was dispersed in DMSO (dimethyl sulfoxide) to form a stock solution of ^-2 Molar concentration. This was followed by a series of dilutions that were made after diluting the stock solution in mili-Q water. For the organ bath experiments, these serial dilutions were incubated further in an in vitro organ bath containing Krebs–Henseleit solution Krebs–Henseleit solution [pH 7.4 (mM: NaCl 118.1, KCl 4.69, KH_2_PO_4_ 1.2, NaHCO_3_ 25.0, D (+) glucose 11.7, MgSO_4_.7H_2_O 1.1, CaCl_2_ 2.5], to obtain the final concentration of a drug [36].

### 2.4. Organ Bath Studies

Contractility studies were conducted on bladder tissue from young rats (7–8 weeks) and older rats (4–5 and 7–9 months) (*n* = 5, each group). An initial force was applied to each strip of tissue by shortening the string resulting in an applied tension (0.5–1 g). The tissue was allowed to stabilise for 60 min and then monitored for spontaneous contractions.

### 2.5. Spontaneous Contractions

After equilibration and initiation of spontaneous contractions, bladder tissue was incubated in an organ bath with different concentrations of oxytocin receptor antagonists, i.e., atosiban (AT) (1 µM and 10 µM), cligosiban (1 µM and 10 µM), and ß-Mercapto-ß, ß-cyclo pentamethylene propionyl (ßMßßC) (1 µM).

### 2.6. Contractile Responses to Oxytocin

Following the spontaneous contractions, response to oxytocin (OT) (10 pM,100 pM,1 nM, 10 nM,100 nM,1 µM) was generated pre- and post-incubation with atosiban (AT) (1 µM and 100 mM), cligosiban (1 µM and 10 µM), and ßMßßC (1 µM) for 10 min at each concentration. To test the viability of tissue, a high concentration of potassium chloride (20 mM) was used before and after the washout phase of the experiment.

The parameters measured during tension recordings were: (i) integral (area under the curve (AUC)), (ii) maximum value, and (iii) frequency [30]. These parameters were first extracted from the data acquisition system and then recorded within a personal computer using Chart Pro v 7.3.8 (ADInstruments, Bella Vista, NSW, Australia). This data was later transferred to a spreadsheet where they were measured as a percentage relative to the high concentration of potassium chloride (KCl; 20 mM). Data was represented as mean ± SEM that denotes standard deviation (mean ± SD), with statistical analysis performed by two-way ANOVA with Tukey’s multiple comparison test, and half-maximal value (EC_50_) calculated using a Graph Pad Prism version 9.0.1 (GraphPad Software, La Jolla, CA, USA)]. * *p* value < 0.05 denotes significance [36].

### 2.7. Immunohistochemistry and Tissue Stains

Tissue slices embedded with paraffin, measuring 4 µm in thickness, were baked for 30 min at 60 °C, dewaxed in 100% xylene for 3 times in a row, and then rehydrated in 100% ethanol 2 more times. The sections were then washed with buffer (1 × DAKO EnVision Flex Wash Buffer) (DAKO, Cat # K8000) twice for 5 min at room temperature, after being rinsed in distilled water. The sections were then incubated for 30 min at 98 °C in DAKO PT Link with a low-pH target retrieval solution (DAKO, Cat # S1699) to extract the epitopes of an antigen. Using a DAKO Autostainer Plus [Ft. Collins, CO, USA], the following procedures were followed for immunostaining: (i) AffiniPure Fab Fragment Donkey Anti-Mouse IgG (H + L) (Jackson ImmunoResearch Labs Inc., West Grove, PA, USA, # 715-007-003) 200 µg/mL diluted in DAKO Protein Block Serum Free (DAKO, Cat # X0909) was used to block the rat-on-mouse reaction for 60 min at room temperature; (ii) the tissue sections were washed with buffer (1 × DAKO EnVision Flex Wash Buffer) twice for 5 min at room temperature; (iii) the tissue sections were incubated with primary antibodies and isotype controls diluted in DAKO Antibody Diluent (DAKO, Cat # S0809) for 60 min at room temperature: Monoclonal Mouse-anti-Smooth Muscle Actin [1:400 dilution (2.5 µg/mL), clone 1A4, DAKO] and Polyclonal Rabbit-anti-Oxytocin Receptor [1:50 dilution (20 µg/mL), Bioss, # B601086491], in addition to Mouse IgG isotype control: Donkey Anti-Mouse IgG (H+L) 1:400 dilution (20 µg/mL) (Jackson ImmunoResearch Labs Inc., Cat # 715-007-003); Rabbit IgG isotype control: Rabbit Polyclonal Antibody IgG Isotype control 1:125 dilution (20 µg/mL) (CST # 3900S). The tissue slices were then washed 3 times in buffer (1 × DAKO EnVision Flex Wash Buffer) for 10 min each time. Next, they were incubated for 60 min at room temperature with a secondary antibody cocktail that had been diluted in DAKO Antibody Diluent (DAKO, Cat # S0809). Following that, the tissue slices were incubated with Jackson ImmunoResearch’s AffiniPure Donkey anti-Mouse AF647 (1:500 dilution, 1.5 µg/mL) and AffiniPure Donkey anti-Rabbit AF488 (1:500 dilution, 1.5 µg/mL) antibodies. The portions were then cleaned with one DAKO EnVision Flex Wash Buffer 3 times for 10 min each. The tissue sections were then counterstained with DAPI (1:10,000 dilution, Sigma Aldrich) for 15 min at room temperature, and then washed for 5 min with distilled water. After applying a 0.3% Sudan Black B in 70% ethanol solution for one minute at room temperature and washing with distilled water for 5 min, the tissue autofluorescence was reduced. Ultimately, a coverslip was affixed to the slides and fastened using Invitrogen’s Prolong Gold antifade reagent [37].

### 2.8. Analysis of IHC Experiments

The tissue slices from the bladder were imaged at the Monash Histology Platform using the VS120 Automated Slide Scanner. DAPI (120 ms) (nuclear staining), AF 488 (1000 ms) (oxytocin receptors), and AF 647 (590 ms) (smooth muscle actin) were the different fluorophores that were utilised for the experimental slides. QuPath-0.3.2 and the Fiji distribution of Image J were used for the analysis of the images after they were taken using OlyVIA (version 2.9.1) [38]. The average and standard error of the average (mean ± SEM) were computed and imported into an Excel document. Prism 9 was utilised to plot the graphs, and a two-tailed unpaired t-test was employed to establish statistical significance following the confirmation of the Gaussian distribution. * *p* value < 0.05 was considered significant.

## 3. Results

The mean ± SEM weights of the 7–8 weeks rats’ bladders were 0.09 ± 0.01 g, 10–12 weeks rats’ bladders 0.13 ± 0.01 g, 4–5 months rats’ bladders 0.12 ± 0.01, and 7–9 months rats’ bladders were 0.16 ± 0.11 g. The average weight of a 7–8 weeks rats was 300–330 g. Similarly, the mean ± SEM weights of 10–12 weeks rats were 404.5 ± 19.9 g, 4–5 months 453.6 ± 6.01 g, and that of 7–9 months rats were 541.8 ± 1.39 g.

### 3.1. Organ Bath Findings

The frequency of spontaneous contractions was significantly reduced with a higher concentration of atosiban (10 µM) in the older (7–9 months) (Figure 1b), but not in young (7–8 weeks), rat (Figure 1d) bladders. On the other hand, atosiban at a lower concentration of 1 µM non-significantly attenuated the frequency (Figure 1a,c) and integral (AUC) (Appendix A) parameters of spontaneous contractions in both age group rats. Furthermore, both low (1 µM) and high concentrations of atosiban (10 µM) reduced the baseline maximum parameter of spontaneous contractions in older (7–9 months) (Appendix A), but not young (7–8 weeks), rats (Appendix A).

Cligosiban significantly decreased the frequency of spontaneous contractions within the bladder of both age group rats.

Bladder contractility was evaluated using the in vitro organ bath technique and tissue was exposed to low and high concentrations (1 µM and 10 µM) of cligosiban. Potassium chloride KCl at a concentration of 20 mM was used to test the viability of tissue, and in response, a forceful contraction was observed. Both low and high concentrations of cligosiban significantly inhibited the frequency of spontaneous contraction within the bladder of older (4–5 months) (Figure 2a,b) and young (7–8 weeks) (Figure 2c,d) rats. Furthermore, cligosiban (1 µM and 10 µM) significantly attenuated the baseline integral (AUC) parameter of spontaneous contractions within the bladder of older (4–5 months) (Figure 3a,b), but not young (7–8 weeks), rats (Figure 3c,d). Moreover, cligosiban at a concentration of 1 µM was capable of significantly reducing the baseline maximum value parameter of spontaneous contractions within the bladder of older (4–5 months) (Appendix A), but not the young (7–8 weeks) (Appendix A), rats. 

Lower concentration (1 µM) of ßMßßC was capable of decreasing the frequency in older (7–9 months) (Appendix A) but not young (Appendix A), while integral (AUC) in both age group (Appendix A) rats. However, no effect was observed on the maximum parameter in both older (7–9 months) ((Appendix A) and young (7–8 weeks) (Appendix A) rats, measured as percentage relative to the maximum percentage of KCl (20 mM).

Oxytocin significantly increased the spontaneous contractions of rat bladder.

Contractility studies were conducted on bladder tissue from young (7–8 weeks) and older (7–9 months) rats (*n* = 5, each group) via a tension gauge organ bath with a cumulative dose–response curve to oxytocin (OT) (10 pM, 100 pM, 1 nM, 10 nM, 100 nM, 1 µM, 10 µM) (Figure 4a,b). Oxytocin (1 µM) significantly increased the bladder contractions within both young and older rats. The only significant effect of oxytocin was seen at its highest concentration, i.e., 1 µM (Figure 4a,b). To evaluate the viability of the tissue, a forceful contraction was observed after incubating the tissue within the organ bath with potassium chloride (KCl; 20 mM), which was done before and after the experiment.

Oxytocin receptor antagonists significantly decreased oxytocin-induced bladder contractions.

OT-induced contractions were significantly inhibited by ßMßßC (1 µM) and atosiban (10 µM) in rats from both age groups. A significant reduction in the % age change in the integral parameter of oxytocin-induced bladder contractions (10 pM, 100 pM, 1 nM, 10 nM, 100 nM, and 1 µM) was observed after incubating the tissue with ßMßßC (1 µM) (Figure 4a,b) and atosiban (10 µM) (Figure 5a,b). This antagonistic effect from ßMßßC (1 µM) and atosiban (10 µM) was observed mainly with the high concentration of OT (1 µM). Furthermore, cligosiban (1 µM and 10 µM) exhibited a non-significant effect on the OT-induced contractions (10 pM, 100 pM, 1 nM, 10 nM, 100 nM, and 1 µM) in both young rats (7–8 weeks) (Appendix A) and older (7–9 months) (Appendix A) rats.

Effects of oxytocin receptor antagonists on the vasopressin-induced bladder contractions.

Vasopressin showed a significant effect on bladder contractions (integral parameter), while no antagonistic effect was detected from oxytocin receptor antagonists, i.e., atosiban (1 µM and 10 µM) in attenuating the bladder contractions in both young (Figure 6a) and older (Figure 6b) rats. However, cligosiban (1 µM and 10 µM) had some effect on the bladder contractions and this effect was achieved at the high concentration of vasopressin (100 nM and 1 µM) in old (Appendix A), but not in young (Appendix A), rats. Furthermore, ßMßßC (1 µM) exhibited a trend in decreasing VP-induced bladder contractions in both young (Appendix A) and old (Appendix A) rats, but this effect was statistically non-significant. All these parameters were measured as a percentage relative to the maximum percentage of KCl (20 mM).

### 3.2. Immunohistochemistry Findings

Oxytocin receptor (OXTR) is significantly expressed within the epithelial and stromal regions of the bladder and is principally nuclear-stained. IHC demonstrated mapping of the oxytocin receptor (OXTR) within the bladder of different age groups of rats (7–8 weeks and 16 weeks). To highlight the colocalization of OXTR with the contractile (smooth muscle) cells, double immunofluorescence staining was conducted for OXTR and α-smooth muscle actin (α-SMA). OXTR expression was detected in both epithelial and smooth muscle (stroma) regions of young [Figure 7B(i–iii)],and older [Figure 7C(a–d)] rats. Graphs indicate that OXTR was located within the cytoplasm and nucleus of epithelial and smooth muscle cells, but was mostly nuclear specific, and predominantly expressed with the smooth muscle cells [Figure 8a (young), and Figure 8b (old)], indicating oxytocin (OT) as a modulator of bladder contractility.

## 4. Discussion

The current study provides insight into how the oxytocin and oxytocin receptor antagonists modulate the spontaneous activity of the bladder, which might assist in the management of LUTS-associated OAB. This study compares the antagonistic effects of the three oxytocin receptor antagonists, namely atosiban, cligosiban, and ßMßßC, on spontaneous as well as oxytocin-induced bladder contractions within young and older male rats. Further, the possibility of cross-signalling between oxytocin and vasopressin was assessed by investigating the in vitro organ bath effects of three oxytocin receptor antagonists (atosiban, cligosiban and ßMßßC) on vasopressin-induced bladder contractions.

Both oxytocin (OT) and arginine vasopressin (AVP) have a similar structure, differing only in the 3rd and 8th positions of amino acid, i.e., isoleucine and leucine at positions 3 and 8 in oxytocin are replaced by phenylalanine and arginine in vasopressin [38]. Thus, because of the similarity in structure, their receptors (OXTR and AVPR) are similar as well [39]. There is only one type of oxytocin (OXTR) and three types of vasopressin receptors (AVPR1A, AVPR1B, and AVPR2) identified, and they are mainly from subfamily A6 of the rhodopsin-type (class I) GPCRs that consist of seven transmembrane helices, three intra and three extracellular loops, an extracellular N-terminus, and an intracellular C-terminus [23,40,41]. OT is mainly associated with functions such as tissue contractility and social bonding while VP mainly deals with water homeostasis and blood pressure regulation. Thus, because of the similarity of these hormones and their receptors, there is a possibility of cross-talk of OT and its antagonists with vasopressin receptor (VPR) that may occur, especially at high concentrations that can result in side effects related to homeostasis such as headache and dizziness [39,42,43]. It was also observed that vasopressin receptor (AVPR1A) might cross-talk with OT in mediating (i) uterine contractions [44]; and (ii) prostatic urethra, bladder neck, and ejaculatory duct contractions [42]. Moreover, numerous studies have also examined the selectivity of OT and AVP and have also revealed the cross-talk between these hormones and their receptors utilising a list of techniques such as immunohistochemistry (IHC) and western blotting (WB) in different species [39,45,46]. Furthermore, it was also reported that AVP has a similar affinity to both OT and AVP receptors, while OT has a higher affinity to OT than to AVP receptors [39,45,46]. Thus, the similarities between OT, VP, and their receptors with their ability to cross-talk emphasise current interest in exploring more selective OT receptor antagonists in attenuating OT-induced bladder contractions [47].

In the current study, cligosiban (1 µM and 10 µM) significantly decreased the frequency of spontaneous contractions within the bladder of both young and older rats (**** *p* < 0.0001). However, a high dose of atosiban (10 µM) significantly reduced the frequency of spontaneous contractions within the bladder of older rats only (** *p* < 0.01). Furthermore, this study also highlighted that oxytocin (OT) significantly increased bladder contractility in both age group rats. However, the effects of oxytocin on bladder contractions are small (% change less than 10% in all parameters), and this effect was observed only with the high concentration of OT (1 µM). However, this indicates the involvement of exogenous oxytocin in maintaining smooth muscle tone within the bladder. Moreover, oxytocin receptor antagonists such as ßMßßC (**** *p* < 0.0001) and atosiban (** *p* < 0.01) are working selectively at OT receptors, and thus are capable of significant inhibition of OT-induced bladder contractions in both young and older rats, supporting the possibility of these oxytocin receptor antagonists as a viable pharmaceutical candidate for the treatment of OAB.

Brading and her colleagues made an initial observation that there is a myogenic basis for OAB. This study then contributed to a significant change in the field of bladder research in (i) understanding the underlying mechanism causing myogenic contractions within the bladder; and (ii) identifying the suitable therapeutics that could be effective in treating this condition [48,49]. These findings were supported by Romine et al. (1985) who evaluated the in vitro organ bath effects of oxytocin on bladder contractions [50,51]. Bladder samples were obtained from rabbits (*n* = 8, male, weight range = 1.8–2.4 kg), and the cumulative response curve to OT was then plotted with various concentrations of 1 nM, 10 nM, 100 nM, 1 µM, and 10 µM. In general, OT at a concentration of 10 µM showed a maximum effect, i.e., 12% increase in maximum contractile amplitude (ED_50_ = 125 nM), compared with the control response generated by 1 µM carbachol. These OT-induced contractions were then tested with antagonists such as phentolamine, atropine, methysergide, saralasin, naloxone, and indomethacin, with only indomethacin (1 µM) showing a partial antagonistic effect on OT (1 µM)-induced contractions. The existence of oxytocin receptors (OXTRs) within the bladder was then confirmed by conducting isotherm binding experiments [50]. Contrary to this, Tarhan et al. (2020) investigated the in vitro organ bath effects on bladder (detrusor) contractions. Bladder samples were taken from men (*n* = 4 male and *n* = 3 female; mean age 63 years, age range 28–80 years) undergoing radical cystectomy for the treatment of invasive bladder carcinoma. Uterine tissues (*n* = 4) obtained from the Wistar rats were used as a control. Contractile responses were tested using carbachol at a concentration of 10 µM in an organ bath to assess the functionality of bladder strips. Dose-response curves to oxytocin (1 nM, 10 nM, 100 nM, 1 µM, and 10 µM) were generated pre- and post-incubation with atosiban (10 µM). Overall, OT caused no change in bladder contractions, with only one strip showing a slight contraction with a very high concentration of 3 × 1 µM. These slight contractions were eliminated by treatment with atosiban, an oxytocin receptor antagonist. Therefore, this study provided supporting rationale to further evaluate the existence of OXTR within the bladder [52].

The present study also observed that the effects of vasopressin on the bladder were greater than oxytocin, though the effects of both were statistically significant. The differences in the magnitude of their effects might be due to the presence of one type of oxytocin (OXTR) and three types of vasopressin receptors (AVPR1A, AVPR1B, and AVPR2) within the bladder [23,40,41]. Moreover, in vitro studies demonstrated that arginine vasopressin can induce bladder contractions in humans, rabbits, and rats due to the existence of arginine vasopressin receptors (AVPR1A, AVPR1B, and AVPR2) in the detrusor muscle [53,54,55,56]. qPCR and WB confirmed the expression of vasopressin receptor subtypes within the bladder of adult female Wistar rats, with AVPR1A > AVPR1B > AVPR2 [53]. Similarly, a series of additional studies also highlighted the existence of these receptors within the bladder. For example, research by Birder et al. (2018) utilised WB to determine the age-mediated effects on the vasopressin receptor expression within the bladder mucosa of female rats. Birder noted that there are two vasopressin receptor types, i.e., AVPR1A and AVPR2 expressed, and with ageing, the expression of AVPR2 significantly increased (* *p* < 0.05) in the bladder mucosa of both age groups (3 months and 25–30 months female Fisher 344 rats; *n* = 8). This age-related increase in AVPR2 receptor expression may also participate in the effectiveness of vasopressin in age-related nocturia [57].

The present study also investigated the possibility of cross-signalling between oxytocin and vasopressin (VP) by investigating the in vitro organ bath effects of oxytocin receptor antagonists (atosiban, cligosiban, and ßMßßC) on vasopressin-induced bladder contractions, respectively. We determined that ßMßßC (1 µM) showed a trend in decreasing VP-induced bladder contractions in both young and older rats but this effect was statistically non-significant. Moreover, cligosiban (1 µM and 10 µM) showed some effect on bladder contractions and this effect was achieved at the high concentrations of vasopressin (100 nM and 1 µM) in older but not young rats. Contrary to this, atosiban (1 µM and 10 µM) had no antagonistic effect in both young and older rats. Furthermore, our research is the first that comments on the distribution of OXTRs within the epithelial and stromal compartments of the bladder. For this purpose, dual staining immunofluorescence (OXTR-oxytocin receptor and α-SMA-alpha smooth muscle actin) was performed and showed that OXTR was mainly located in the nucleus of a cell and this intensity of staining was greater in the stromal part of the bladder. Moreover, these receptors were colocalised with the smooth muscle cells, indicating oxytocin (OT) as a modulator of bladder contractility in young and older rats. These findings were supported by Cafarchio et al. (2020), who quantitatively confirmed the existence of mRNA for OXTR within the bladder (14–16 week-old female Wistar rats; *n* = 32) using the PCR technique, highlighting its local effect via decreased intravesical pressure and thus used to reduce tension within the bladder [31]. On the other hand, Williams (2002) reviewed the distribution of muscarinic receptors within the bladder. He concluded that these receptors are normally present within the bladder at three different locations, i.e., urothelium (epithelial lining), detrusor muscle, and prejunctionally at the cholinergic and adrenergic nerve terminals. However, it is still unknown which specific subtype of these muscarinic receptors is involved in the pathophysiology of OAB, which needs to be addressed [58]. Therefore, based on current findings and those found in the existing literature, it is important to further examine the microenvironment within the bladder to better understand its role in the development of OAB.

## 5. Conclusions

Overall, OAB is a highly prevalent condition in middle and older age populations with limited pharmacotherapy available. Our data demonstrated the importance of oxytocin (OT) as a paracrine modulator of bladder contractions and the use of oxytocin receptor antagonists in attenuating spontaneous and oxytocin-induced bladder contractions in different age groups in male rats. In addition, due to time constraints, this study did not further investigate the human overactive bladder, which is considered one of the limitations of the current study. Therefore, further studies are required to determine the biological role of these oxytocin receptors in OAB to improve our understanding of OAB and LUTS and to explore future therapeutic opportunities.

## Figures and Tables

**Figure 1 biomedicines-12-00674-f001:**
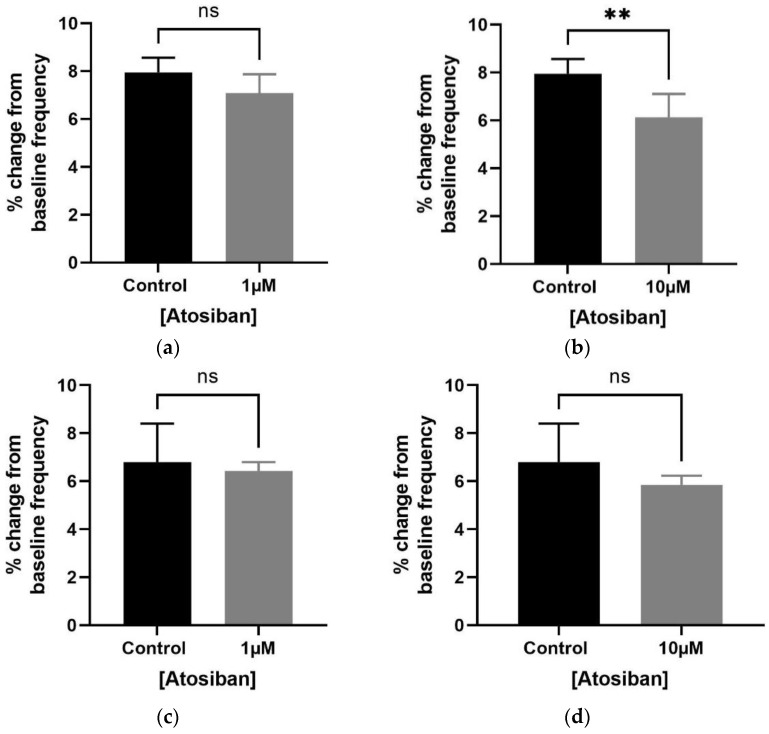
Exogenous atosiban attenuated the myogenic tone by significantly reducing the frequency of bladder (spontaneous) contractions in older rats. To evaluate the myogenic tone, rat bladder was exposed to atosiban (10 µM), resulting in a significant % change in the frequency of bladder contractions within the bladder of older (7–9 months) (**d**), but not young (7–8 weeks), (**b**) rats. Atosiban at a concentration of 1 µM also non-significantly decreased the frequency of bladder contractions [young (**a**) and old (**c**)] [unpaired *t*-test, *n* = 5, ** *p* < 0.01] (** shows significant effects while ns indicates non-significant effects).

**Figure 2 biomedicines-12-00674-f002:**
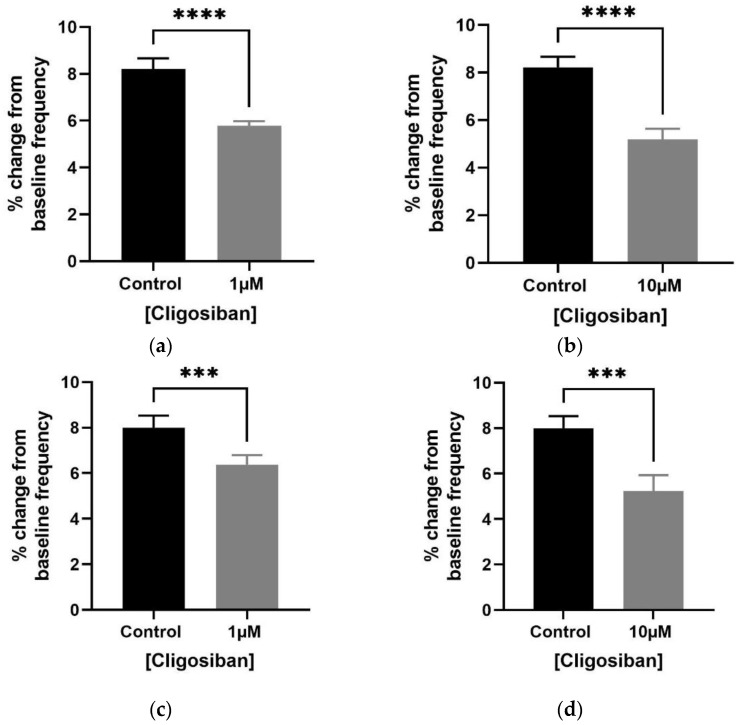
Exogenous cligosiban (1 µM and 10 µM) significantly downregulated the frequency of spontaneous contractions within both older (4–5 months) (**** *p* < 0.0001) and young (7–8 weeks) (**a**,**b**) (*** *p* < 0.001) rats’ bladders (**c**,**d**) (**** *p* < 0.0001) [unpaired *t*-test, *n* = 5] (**** and *** shows significant effects).

**Figure 3 biomedicines-12-00674-f003:**
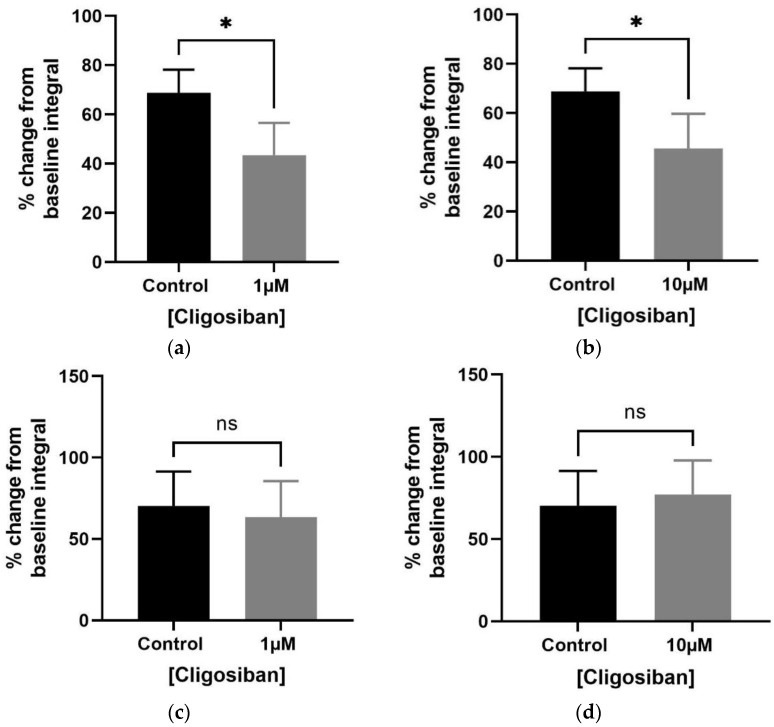
Cligosiban (1 µM and 10 µM) altered the smooth muscle tone by significantly decreasing the integral (AUC) parameter of bladder (spontaneous) contractions within the bladder of older (4–5 months) (**a**,**b**), but not young (7–8 weeks), rats (**c**,**d**), represented as a percentage change measured relative to the maximum percentage of KCl (20 mM) [unpaired *t*-test, *n* = 5, * *p* < 0.05] (* indicates significant effects while ns shows non-significant effects).

**Figure 4 biomedicines-12-00674-f004:**
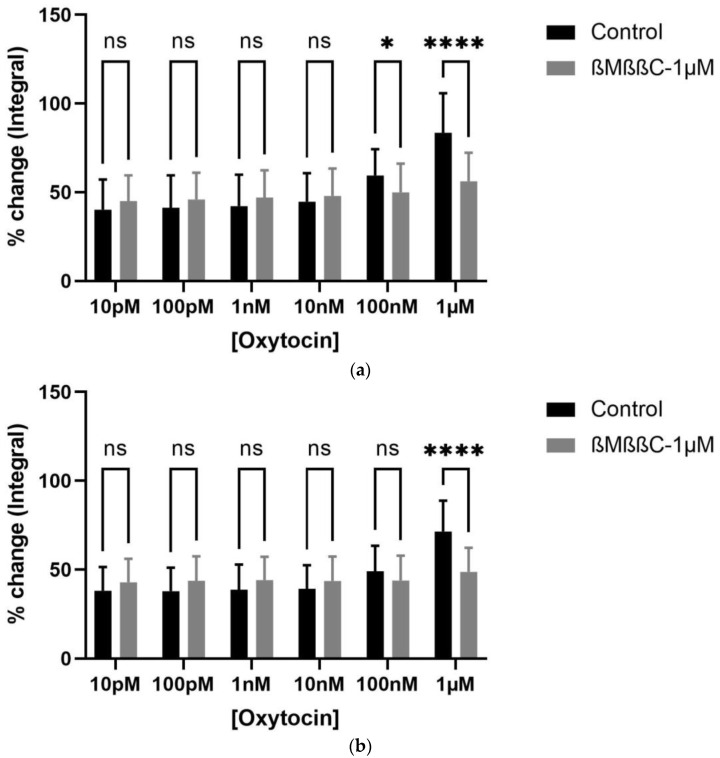
Age-mediated effects of exogenous oxytocin (1 µM) on the myogenic tone of the bladder by significantly increasing bladder contractility. Application of ßMßßC (1 µM) oxytocin receptor antagonist significantly decreased the integral (AUC) parameter of oxytocin-induced bladder contractions (1 µM) in 7–8 weeks (young) (**a**) and 7–9 months (older) (**b**) rats, respectively, calculated as a percentage change of the integral (AUC) that is measured relative to the maximum percentage of KCl (20 mM) [two-way ANOVA Sidak’s multiple comparisons test, * *p* < 0.05 and **** *p* < 0.0001] (* and **** shows significant effects while ns denotes non-significant effects).

**Figure 5 biomedicines-12-00674-f005:**
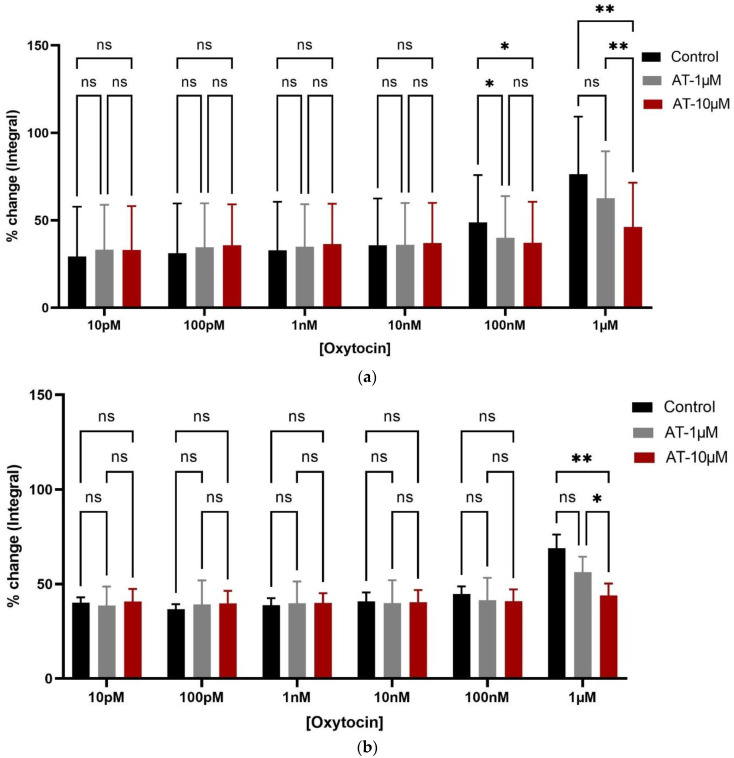
Effects of atosiban on bladder contractions. Atosiban (10 µM) has a significant antagonistic effect in decreasing the oxytocin (OT, 1 µM)-induced bladder contractions in both young (7–8 weeks) (**a**) and older (7–9 months) (**b**) rats, measured as a percentage of change in integral (AUC) that is calculated relative to the maximum percentage of KCl (20 mM) [two-way ANOVA Tukey’s multiple comparison test, * *p* < 0.05 and ** *p* < 0.01] (* and ** shows significant effects while ns indicates non-significant effects).

**Figure 6 biomedicines-12-00674-f006:**
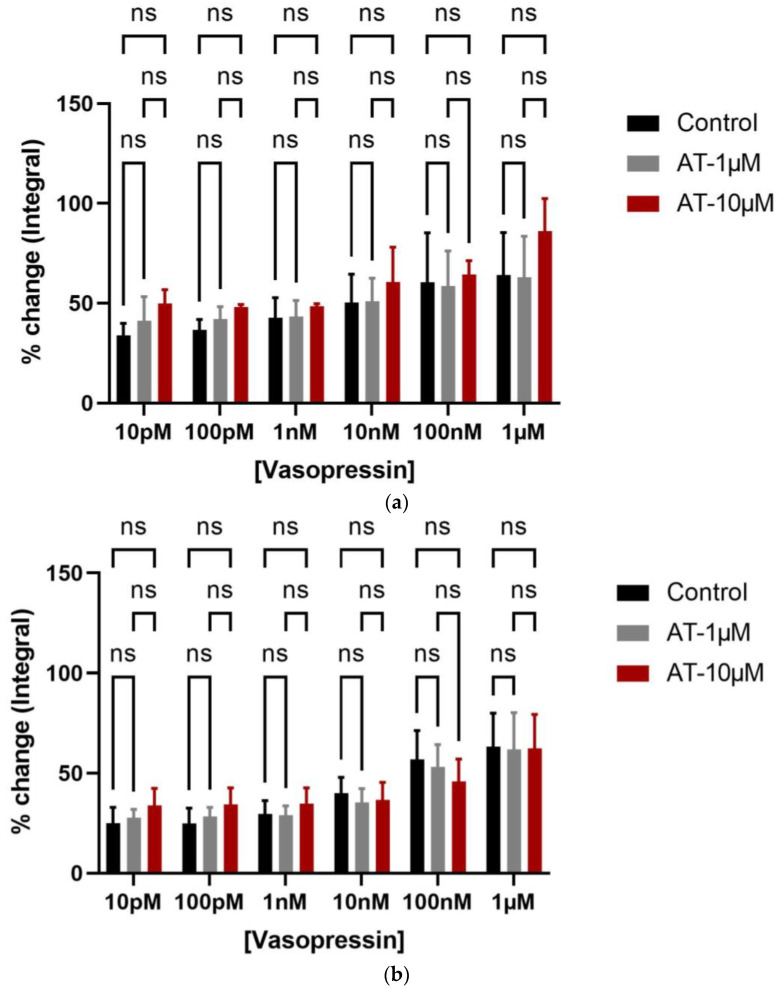
Effects of exogenous vasopressin and atosiban on the bladder movements. Vasopressin (1 µM and 100 nM) significantly increased bladder contractions, while atosiban (1 µM and 10 µM) did not affect the % change in the integral parameter of vasopressin-induced bladder contractions in both young (**a**) and older (**b**) rats [two-way ANOVA Tukey’s multiple comparison test, *p* ≥ 0.05] (ns denotes non-significant effect).

**Figure 7 biomedicines-12-00674-f007:**
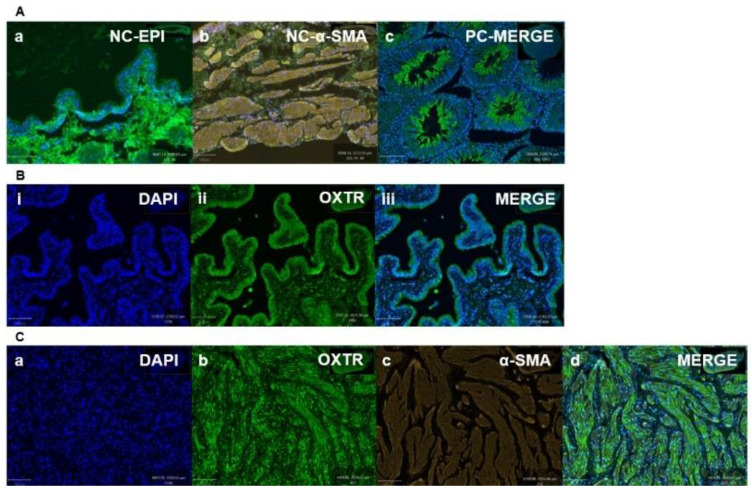
Mapping of OXTR in the young and older rats’ bladders. Illustrates the immunohistochemistry (IHC) staining of negative control for the oxytocin receptor in epithelial (**A**(**a**)) and stromal (**A**(**b**)) regions, respectively; and positive control (rat testis) (**A**(**c**)) for oxytocin receptor. Images (**B**(**i**),**C**(**a**)) DAPI reveal (nuclear) staining; (**B**(**ii**),**C**(**b**)) demonstrate OXTR staining; (**C**(**c**)) indicates actin staining; and the (**B**(**iii**),**C**(**d**)) merged images represent colocalization of OXTR with epithelial and contractile cells (actin), respectively (scale bar = 100 µm) [NC-EPI = Negative Control Epithelium; NC- α-SMA = Negative Control Smooth Muscle; PC-MERGE = Positive Control Merge (Testis); DAPI = 4’,6-diamidino-2-phenylindole; OXTR = Oxytocin Receptor; α-SMA = Alpha Smooth Muscle Actin; MERGE = Merged Image].

**Figure 8 biomedicines-12-00674-f008:**
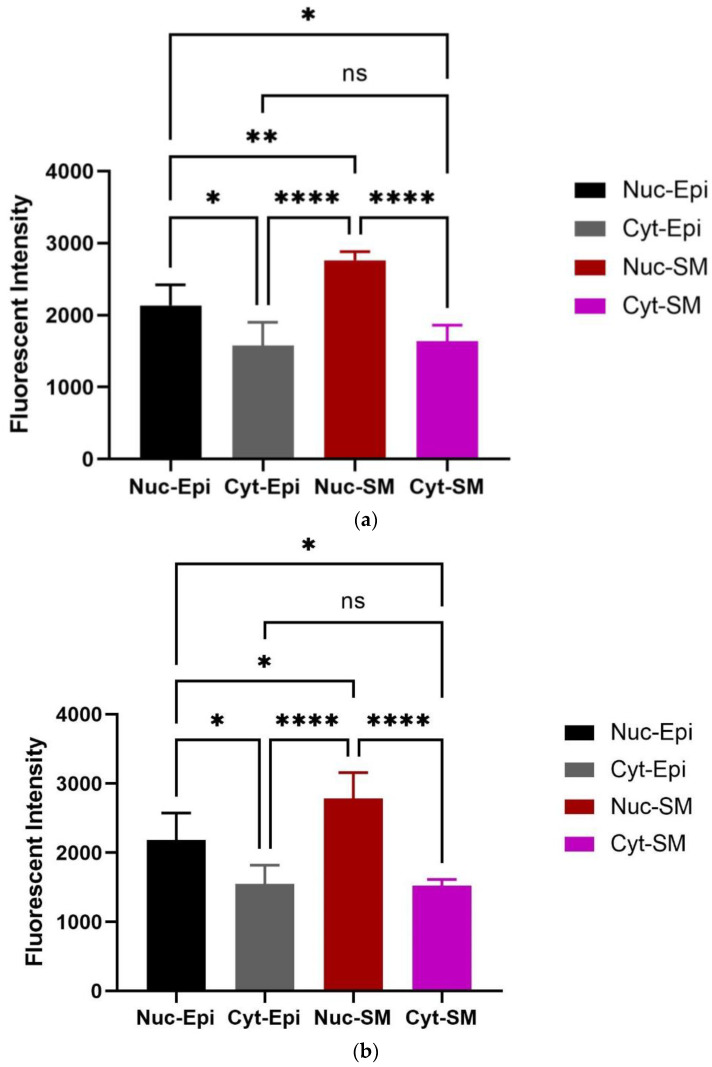
Qualitative analysis of oxytocin receptor (OXTR) within the bladder. Graphs (**a**,**b**) demonstrate that OXTR is located within the cytoplasm and nucleus of both epithelial and smooth muscle cells, but is mostly nuclear specific, and is predominantly expressed in the smooth muscle cells of both young (7–8 weeks) and old (16 weeks) rats (One-way ANOVA with Tukey’s multiple comparison test, *n* = 5, * *p* < 0.05, ** *p* < 0.01 and **** *p* < 0.0001) [Nuc-Epi = Nucleus Epithelium; Cyt-Epi = Cytoplasm Epithelium; Nuc-SM = Nucleus Smooth Muscle; and Cyt-SM = Cytoplasm Smooth Muscle] (*, ** and **** indicates significant effects while ns shows non-significant effects).

## Data Availability

Data is unavailable due to privacy or ethical restrictions.

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
