# Peer review of "Age-Dependent Effects of Oxytocin and Oxytocin Receptor Antagonists on Bladder Contractions: Implications for the Treatment of Overactive Bladder Syndrome"

_biomedicines, 2024, doi:10.3390/biomedicines12030674_

Round 1

Reviewer 1 Report

Comments and Suggestions for Authors

This manuscript reported the organ bath bladder contractility to oxytocin and it's antagonist, immunohistochemistry study of oxytocin receptors,  and the potential treatment target to overactive bladder (OAB) in young and old rats. This study is interesting and clinically relevant. There are several critical points that the authors should address:

1) This is an animal study. The authors should comment on the limitation of transferring the results f this study to human overactive bladder.

2) OAB is multifactorial, ageing is one of the pathophysiological factors for OAB. The authors should check the changes of muscarinic receptors and the contractility to carbachol and compare the bladder contractility to oxytocin and carbachol, which might provide evidence for the ratio of oxytocin  to carbachol dependent contractility.

3) Searching for novel pharmacological target to OAB is worthy congratulations. However, the authors should further address the role of oxytocin receptors, muscarinic receptors, and beta-3 adrenoceptors in the pathophysiology of OAB.

Author Response

Reviewer 1

Comments

Responses

1) This is an animal study. The authors should comment on the limitation of transferring the results of this study to human overactive bladder.

Thank you, suggested change has been incorporated into the revised manuscript.  Changes in response to this are highlighted yellow and made in Line number 478-480 page 16 in the Conclusions Section of the revised manuscript.

2) OAB is multifactorial, ageing is one of the pathophysiological factors for OAB. The authors should check the changes of muscarinic receptors and the contractility to carbachol and compare the bladder contractility to oxytocin and carbachol, which might provide evidence for the ratio of oxytocin to carbachol-dependent contractility.

Thank you, suggested change has been incorporated into the revised manuscript.  Changes in response to this are highlighted yellow and made in Line number 464-470 pages 15-16 in the Discussion Section of the revised manuscript.

3) Searching for the novel pharmacological target to OAB is worthy congratulations. However, the authors should further address the role of oxytocin receptors, muscarinic receptors, and beta-3 adrenoceptors in the pathophysiology of OAB.

Thank you, suggested change has been incorporated into the revised manuscript.  Changes in response to this are highlighted yellow and made in Line numbers 57-72 and 95-97 page 2 in the Introduction Section of the revised manuscript.

Reviewer 2 Report

Comments and Suggestions for Authors

The article has a very interesting topic. It is written in a very neat way, but unfortunately it has an omission. The Material and Methods section doesn't contain the description of vasopressin-induced bladder contractions section of the study. In the Results section it is described how vasopressin showed a significant effect on bladder contractions and the way oxytocin receptor antagonists have or not an antagonistic effect on it but the way this was evaluated is not previously described (in fact the word vasopressin does not exist in the Material and Methods section) - please review and add the relevant informations.

The Conclusions section should include only data generated by the present study - please review.

Author Response

Reviewer 2

1)     The Material and Methods section doesn't contain the description of vasopressin-induced bladder contractions section of the study. In the Results section it is described how vasopressin showed a significant effect on bladder contractions and the way oxytocin receptor antagonists have or not an antagonistic effect on it but the way this was evaluated is not previously described (in fact the word vasopressin does not exist in the Material and Methods section) - please review and add the relevant informations.

Thank you, suggested change has been incorporated into the revised manuscript.  Changes in response to this are highlighted yellow and made in Line numbers 120-121 page 3 in the Materials and Methods Section of the revised manuscript.

2)     The Conclusions section should include only data generated by the present study - please review.

Thank you, suggested change has been incorporated into the revised manuscript.  Changes in response to this are highlighted yellow and green made in Line numbers 474-482 page 16 in the Conclusions Section of the revised manuscript.

Round 2

Reviewer 1 Report

Comments and Suggestions for Authors

The authors have revised the manuscript and addressed all points in the revised version. This manuscript is currently acceptable for publication in this journal.

Author Response

All the comments have already been addressed. No further action required.

Reviewer 2 Report

Comments and Suggestions for Authors

The authors stated "Thank you, suggested change has been incorporated into the revised manuscript. Changes in response to this are highlighted yellow and made in Line numbers 120-121 page 3 in the Materials and Methods Section of the revised manuscript."

However, no highlighted yellow modifications are present in Materials and Methods section. Please review.

All the other issues were adressed in a satisfactory manner.

Author Response

The authors stated "Thank you, suggested change has been incorporated into the revised manuscript. Changes in response to this are highlighted yellow and made in Line numbers 120-121 page 3 in the Materials and Methods Section of the revised manuscript."

However, no highlighted yellow modifications are present in Materials and Methods section. Please review.

All the other issues were addressed in a satisfactory manner.

Thank you, suggested change has been incorporated into the revised manuscript.  Changes in response to this are highlighted yellow and made in Line numbers 121 page 3 in the Materials and Methods Section of the revised manuscript.

Round 3

Reviewer 2 Report

Comments and Suggestions for Authors

The queries were answered in a satisfactory manner.